# Emetine blocks DNA replication via proteosynthesis inhibition not by targeting Okazaki fragments

David Lukac, Zuzana Machacova, Pavel Moudry (ORCID)

**DNA synthesis of the leading and lagging strands works independently and cells tolerate single-stranded DNA generated during strand uncoupling if it is protected by RPA molecules. Natural alkaloid emetine is used as a specific inhibitor of lagging strand synthesis, uncoupling leading and lagging strand replication. Here, by analysis of lagging strand synthesis inhibitors, we show that despite emetine completely inhibiting DNA replication: it does not induce the generation of single-stranded DNA and chromatin-bound RPA32 (CB-RPA32). In line with this, emetine does not activate the replication checkpoint nor DNA damage response. Emetine is also an inhibitor of proteosynthesis and ongoing proteosynthesis is essential for the accurate replication of DNA. Mechanistically, we demonstrate that the acute block of proteosynthesis by emetine temporally precedes its effects on DNA replication. Thus, our results are consistent with the hypothesis that emetine affects DNA replication by proteosynthesis inhibition. Emetine and mild POLA1 inhibition prevent S-phase poly(ADP-ribosyl)ation. Collectively, our study reveals that emetine is not a specific lagging strand synthesis inhibitor with implications for its use in molecular biology.**

## Introduction

Emetine (EME), an alkaloid present in the plant *Carapichea ipecacuanha*, is the main active ingredient in ipecac syrup which has been used in traditional medicine as an emetic, expectorant and antiparasitic drug. However, its medical use is limited mainly due to both myopathy and cardiotoxicity which are associated with chronic usage of EME (Wang & Yang, 2020). The inhibition of protein, DNA, and RNA synthesis is EME's main mechanism of action in biological systems (Grollman, 1968; Gupta & Siminovitch, 1976). In recent molecular biology research, EME was used, based on an earlier report by Burhans and colleagues (Burhans et al, 1991), as a specific inhibitor of lagging strand synthesis which prevents the formation of Okazaki fragments (OFs), uncoupling leading and lagging strand replication (Hanzlikova et al, 2018; Thakar et al, 2020; Xiao et al, 2020; Cong et al, 2021; Yamashita et al, 2022).

DNA replication occurs in opposite directions on each of the two parental strands. DNA replication is initiated on both strands by the synthesis of hybrid RNA/DNA primer by the polymerase $\alpha$ complex (Burgers, 2009), which consists of the catalytic subunit POLA1, accessory subunit POLA2, and primase subunits PRIM1 and PRIM2. Hybrid RNA/DNA primers then become substrates for elongating polymerases $\varepsilon$ and $\delta$ on leading and lagging strands, respectively. DNA synthesis at leading and lagging strands works independently and can be uncoupled in vivo (Ercilla et al, 2020). Strand uncoupling induced by POLA1 inhibition during DNA replication generates single-stranded DNA (ssDNA) that is tolerated by cells if it is protected by the surplus of RPA molecules. DNA synthesis on the leading strand requires only one primer and has mostly uninterrupted DNA polymerase activity. Lagging strand synthesis occurs in a discontinuous manner due to the movement of polymerase in the opposite direction of the replication fork and therefore requires the use of multiple RNA/DNA primers. Pol $\delta$ synthetizes 100–500 nucleotides long Okazaki fragments before it displaces the 5'-end of the preceding primer. OFs are processed by nucleases FEN1 and DNA2 and finally sealed by the action of DNA ligase 1 (LIG1) to form a continuous lagging DNA strand (Burgers & Kunkel, 2017). Unligated OFs that escape canonical processing are sensed by poly(ADP-ribose) polymerase 1 (PARP1) to initiate a backup pathway using XRCC1 and LIG3 (Hanzlikova et al, 2018). PARPs are a family of enzymes that play role in diverse cellular processes and modify themselves and other proteins with mono- or poly(ADP-ribose) (PAR). The most abundant enzyme in the PARP family is PARP1. PARP1 is activated by ssDNA breaks and various DNA replication intermediates including stalled or collapsed replication forks (Sugimura et al, 2008; Bryant et al, 2009; Chaudhuri & Nussenzweig, 2017; Hanzlikova & Caldecott, 2019). We recently implicated PARP in the regulation of the speed of DNA replication. Inhibition of PARP does not slow or block DNA replication but accelerates fork progression during unperturbed replication (Maya-Mendoza et al, 2018). However, the mechanism by which PARP affects fork speed is poorly understood. Given that the PAR signal in unperturbed cells is detected exclusively in S-phase cells, the poly(ADP-ribosyl)ation (PARylation) signal localizes with sites of DNA replication and is associated with unprocessed OFs (Hanzlikova et al, 2018), and PARP inhibition generates ssDNA gaps (Cong et al, 2021), increased

Institute of Molecular and Translational Medicine, Faculty of Medicine and Dentistry, Palacky University, Olomouc, Czech Republic

Correspondence: pavel.moudry@upol.cz

replication fork speed induced by PARP inhibition in the absence of exogenous DNA damage could be the result of unprocessed OFs. Here, by addressing this hypothesis using inhibitors of lagging strand synthesis we found that emetine is not a specific inhibitor of lagging strand synthesis with implications for emetine use in molecular biology.

# Results

### Emetine rapidly and fully impairs DNA replication

To explore the mechanism behind fast replication forks after PARP inhibition, we analyzed replication fork progression after combined

PARP and OF synthesis inhibition. We used EME, an inhibitor of DNA replication that prevents the synthesis of OFs (Burhans et al, 1991). As found previously (Hanzlikova et al, 2018), EME completely suppressed the appearance of PAR induced by PARG inhibition (PARGi) or LIG1 knockdown (Figs S1A–C). Next, using DNA combing assays that monitor replication fork progression by incorporating nucleoside analogs into nascent DNA strands, we confirmed that U2OS cells when treated at 10 $\mu$M PARPi for 16 h (Maya-Mendoza et al, 2018) displayed an increased replication fork rate (Fig 1A). Unexpectedly, 20 min of EME treatment caused a massive reduction in replication fork rate both in control and PARPi treated cells (Fig 1A). These findings indicate that EME inhibits DNA synthesis of both the leading and lagging strands. To further assess whether DNA replication was globally affected by EME, we monitored the

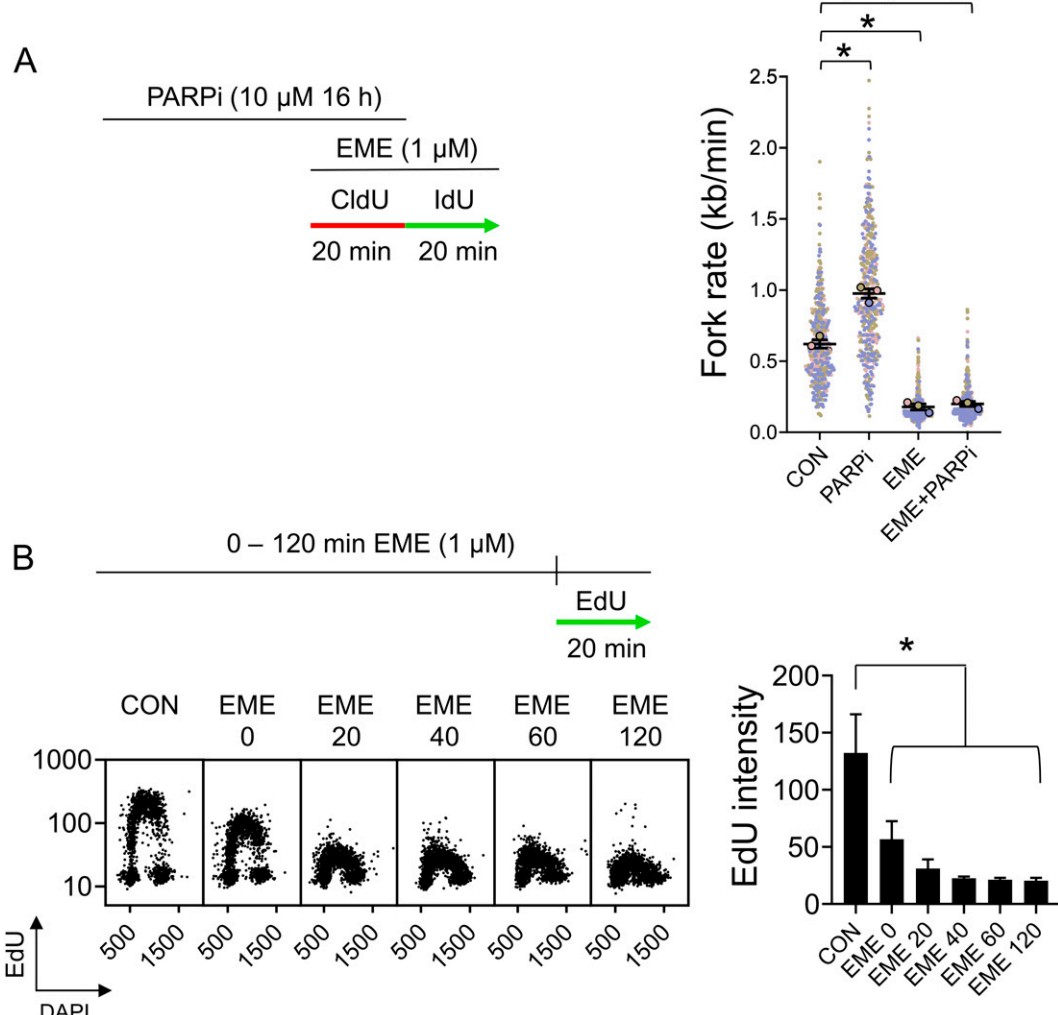

**Figure 1.   Emetine rapidly and fully impairs DNA replication.**
**(A)** U2OS cells were treated with 10 $\mu$M PARP inhibitor Olaparib for 16 h, 1 $\mu$M EME for 40 min, labeled with CldU/IdU, and fork rates were determined by DNA combing according to the length of IdU track. The superplot shows results from three independent experiments (n = 3, blue, pink and brown). Circles show mean values of each experiment containing 150 DNA fibers. The average of mean values from each experiment (black horizontal bar) with standard error of the mean (black error bars) are indicated. Mean values from each experiment were statistically tested by paired two-tailed $t$ test. *$P$ < 0.05. **(B)** U2OS cells were pretreated with 1 $\mu$M EME for the indicated time, treated with 10 $\mu$M EdU for an additional 20 min, and stained for incorporated EdU by click chemistry. The bar graph shows mean EdU intensities ± SD from three independent experiments (n = 3). Statistical analysis by one-way ANOVA. *$P$ < 0.05.

incorporation of EdU into nascent DNA after EME treatment. Simultaneous addition of EME and EdU caused a dramatic reduction in DNA synthesis (Fig 1B). After 20 min of EME pretreatment, DNA synthesis was already fully inhibited (Fig 1B) in the U2OS cell line. These effects of EME on DNA replication were also recapitulated in human diploid RPE1 cells (Fig S1D). In summary, these findings show that emetine rapidly and fully impairs DNA replication.

## Emetine does not cause RPA32 chromatin-loading or generation of ssDNA

The direct consequence of inhibition of OF synthesis is the uncoupling of leading and lagging strands, which is demonstrated by the accumulation of ssDNA and RPA molecules at active forks (Ercilla et al, 2020). To test whether EME induces strand uncoupling, we used QIBC to measure the accumulation of ssDNA and CB-RPA32

in single cells (Toledo et al, 2013). In those assays, we compared the effects of EME with adarotene (ADA), a POLA1 inhibitor that induces strand uncoupling (Ercilla et al, 2020). In contrast to ADA treatment, EME did not induce the accumulation of RPA on chromatin (Figs 2A and S2A), nor the generation of ssDNA (Figs 2B and S2B) in either U2OS or RPE1 cells. An important criterion of strand uncoupling is that RPA-coated ssDNA accumulation and DNA synthesis should occur simultaneously. Indeed, cells treated with ADA for 40 min still incorporated EdU, whereas EME treatment completely inhibited EdU incorporation (Fig S2C and D). This was also observed at the level of individual forks, as the progression of most forks was only slightly reduced during conditions of strand uncoupling induced by ADA (Figs 2C and S2E), despite the simultaneous buildup of RPA-coated ssDNA. In sharp contrast to this, EME caused a massive reduction in replication fork rate (Figs 2C and S2E), suggesting that replication on both DNA strands was dramatically reduced.

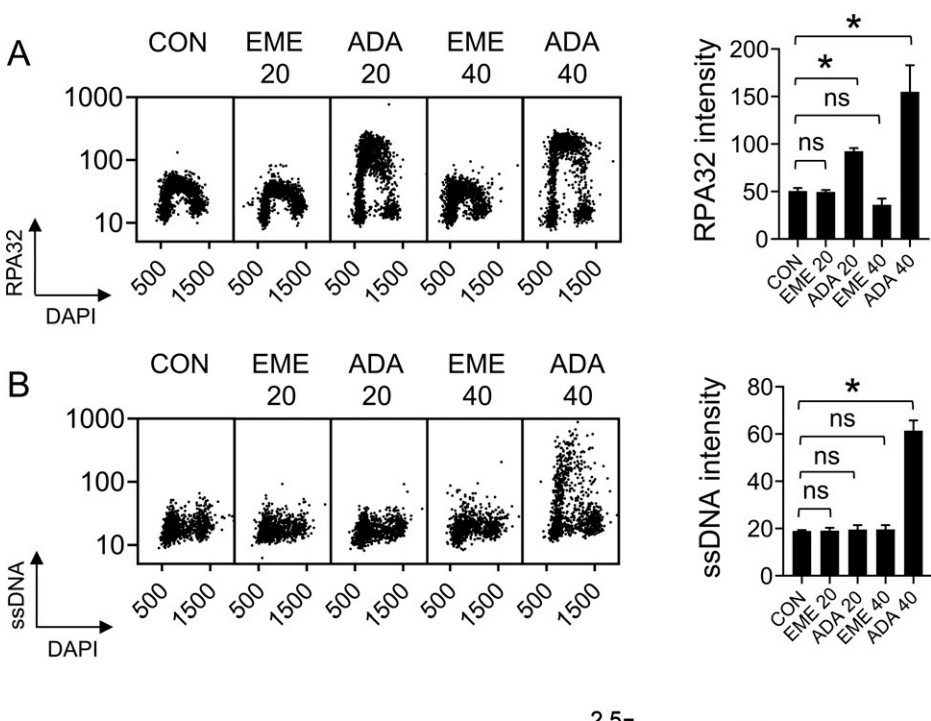

**Figure 2. Emetine does not cause RPA32 chromatin-loading or generation of single-stranded DNA (ssDNA).**
**(A)** U2OS cells were treated with 1 µM EME or 1 µM ADA for 20 or 40 min and stained for CB-RPA32. The bar graph shows mean CB-RPA32 intensities ± SD from three independent experiments (n = 3). Statistical analysis by one-way ANOVA. *P < 0.05; ns, nonsignificant. **(B)** U2OS cells were pretreated with 10 µM BrdU for 48 h, treated with 1 µM EME or 1 µM ADA for 20 or 40 min, and stained for ssDNA using anti-BrdU antibody under non-denaturing conditions. The bar graph shows mean ssDNA intensities ± SD from three independent experiments (n = 3). Statistical analysis by one-way ANOVA. *P < 0.05; ns, nonsignificant. **(C)** U2OS cells were pretreated with 1 µM EME or 1 µM ADA for 20 min, labeled with CldU/IdU, and fork rates were determined by DNA combing according to the length of IdU track. The superplot shows results from three independent experiments (n = 3, blue, pink and brown). Circles show mean values of each experiment containing 150 DNA fibers. The average of mean values from each experiment (black horizontal bar) with standard error of the mean (black error bars) are indicated. Mean values from each experiment were statistically tested by paired two-tailed t test. *P < 0.05; ns, nonsignificant.

## Emetine does not activate the replication checkpoint or DNA damage response

Our findings established that EME does not induce markers of strand uncoupling despite having a dramatic effect on DNA replication. To explore the replication stress response activated by EME treatment, we continued by investigating the activation of DNA damage signaling. In contrast to ADA or HU treatment, we found that EME did not cause DNA damage as measured by γH2AX induction in both U2OS and RPE1 cells (Figs 3A and S3A). In line with that, EME treatment did not lead to the activation of the replication checkpoint as documented by the absence of phosphorylations of H2AX, Chk1, or RPA32 (Fig 3B). This strongly contrasts with the stress responses of replication forks to strand uncoupling (ADA) or to dNTP deprivation (HU), both of which induce phosphorylation of H2AX, Chk1, and RPA32. Moreover, EME pretreatment suppressed ADA-induced phosphorylation of H2AX and accumulation of RPA32 on chromatin (Fig 3C). This result does not reflect the nonspecific effect of EME on DNA damage signaling, as EME did not block UV-induced phosphorylation of H2AX and accumulation of RPA32 on chromatin (Fig S3B). Collectively, our data show that EME does not generate DNA damage nor activate the replication checkpoint; this supports the hypothesis that EME affects DNA replication via a different mechanism than the inhibition of lagging strand synthesis.

## Emetine's anti-proteosynthetic activity blocks DNA replication

To explain our results, we next considered the possibility that EME blocks DNA replication via the inhibition of proteosynthesis. Indeed, EME is a potent inhibitor of protein biosynthesis (Grollman, 1966). Ongoing protein biosynthesis is required for flawless DNA replication; however, the inhibition of protein biosynthesis does not result in ssDNA formation, checkpoint activation, or DNA damage (Mejlvang et al, 2014; Henriksson et al, 2018). To start with, we compared the effects of EME with an inhibitor of proteosynthesis cycloheximide (CHX) on nascent protein synthesis using the O-propargyl-puromycin (OPP) assay. In this assay, OPP is incorporated into newly translated proteins and conjugated with fluorescent dye using a click reaction. We confirmed that both EME and CHX effectively blocked protein biosynthesis (Figs 4A and S4A). More importantly, both EME and CHX also dramatically inhibited DNA replication, as illustrated by EdU incorporation (Figs 4A and S4A) and the progression of individual replication forks (Figs 4B and S4B), which supports the hypothesis that ongoing proteosynthesis is required for DNA replication. To further validate that the inhibition of proteosynthesis is responsible for the effects EME has on DNA replication, we reasoned that the effect on proteosynthesis caused by EME should precede its effects on DNA replication. Indeed, short incubation times revealed that both EME and CHX first block proteosynthesis, followed by inhibition of DNA replication at later time points (Figs 4C and S4C). Collectively, this evidence confirms the notion that EME affects the replication of both DNA strands most likely via the inhibition of protein biosynthesis.

## PARP activity during the S-phase is prevented by mild inhibition of POLA1

Emetine was recently used to establish that unprocessed OFs are the source of S-phase PARylation (Hanzlikova et al, 2018). Since our data indicate that EME is not a specific lagging strand synthesis inhibitor, we asked whether the inhibition of lagging strand synthesis by POLA1 inhibitors ADA or CD437 suppresses S-phase PARylation. In contrast to EME, short incubation with 1 μM ADA or 2 μM CD437 did not suppress PARGi-induced PARylation (Fig 5A). Surprisingly, 1 μM ADA treatment alone led to an increase in PARylation levels (Fig 5A). We noted that 1 μM ADA also strongly induces DNA damage marker γH2AX (Fig 5B) and therefore the increased PARylation signal induced by acute POLA1 inhibition likely reflects a secondary response to extensive DNA damage leading to the replication catastrophe. Consistent with CD437 being a less potent POLA1 inhibitor than ADA (Ercilla et al, 2020), a reproducible but less pronounced increase in PARylation and γH2AX signals was observed also after treatment with 2 μM CD437 (Fig 5A and B). Acute POLA1 inhibition is therefore unsuitable for testing whether unprocessed OFs are the source of S-phase PARylation. We, therefore, used mild conditions for POLA1 inhibition (using 100 nM ADA and 200 nM CD437 overnight) that reduce EdU incorporation (Fig S5A), induce the formation of ssDNA and accumulation of RPA32 at chromatin (Fig S5B and C) and are tolerated by U2OS cells (Fig S5D). Mild POLA1 inhibition suppressed PARGi-induced S-phase PAR signal to similar levels as EME treatment without inducing DNA damage (Fig 5C–E). In addition, mild POLA1 inhibition also partially suppressed PARylation induced by LIG1 knockdown (Fig 5F), further supporting the model where unligated OFs contribute to S-phase PARylation.

# Discussion

Numerous recent reports (Hanzlikova et al, 2018; Thakar et al, 2020; Xiao et al, 2020; Cong et al, 2021; Yamashita et al, 2022) have used emetine as a specific inhibitor of lagging strand synthesis based on the work by Burhans (Burhans et al, 1991). However, little is known about the potential underlying mechanism of EME towards DNA replication in general and lagging strand synthesis in particular. In this study, we confirm earlier results and observe that EME blocks both protein and DNA synthesis. Specifically, we report that EME inhibits DNA replication on both strands and therefore should not be considered as a specific inhibitor of lagging strand synthesis.

Inhibition of lagging strand synthesis results in strand uncoupling caused by the forward movement of replicative helicases together with the leading strand polymerase while DNA synthesis at the lagging strand has stalled. Such experimental conditions should fulfill several criteria. First, the rate of total DNA replication should be reduced to ~50%. In contrast, EME completely blocked DNA replication on both the individual forks and on a global level measured by DNA combing and EdU incorporation, respectively. The second criterion for the inhibition of lagging strand synthesis is the accumulation of ssDNA and RPA protein on the lagging strand. To examine this, we observed ssDNA generation via BrdU detection under non-denaturing conditions and the accumulation of CB-RPA32. EME, in contrast to the POLA1 inhibitor ADA, did not increase either ssDNA or RPA on chromatin. Furthermore, EME even prevented the ADA-induced increase in CB-RPA32 and γH2AX,

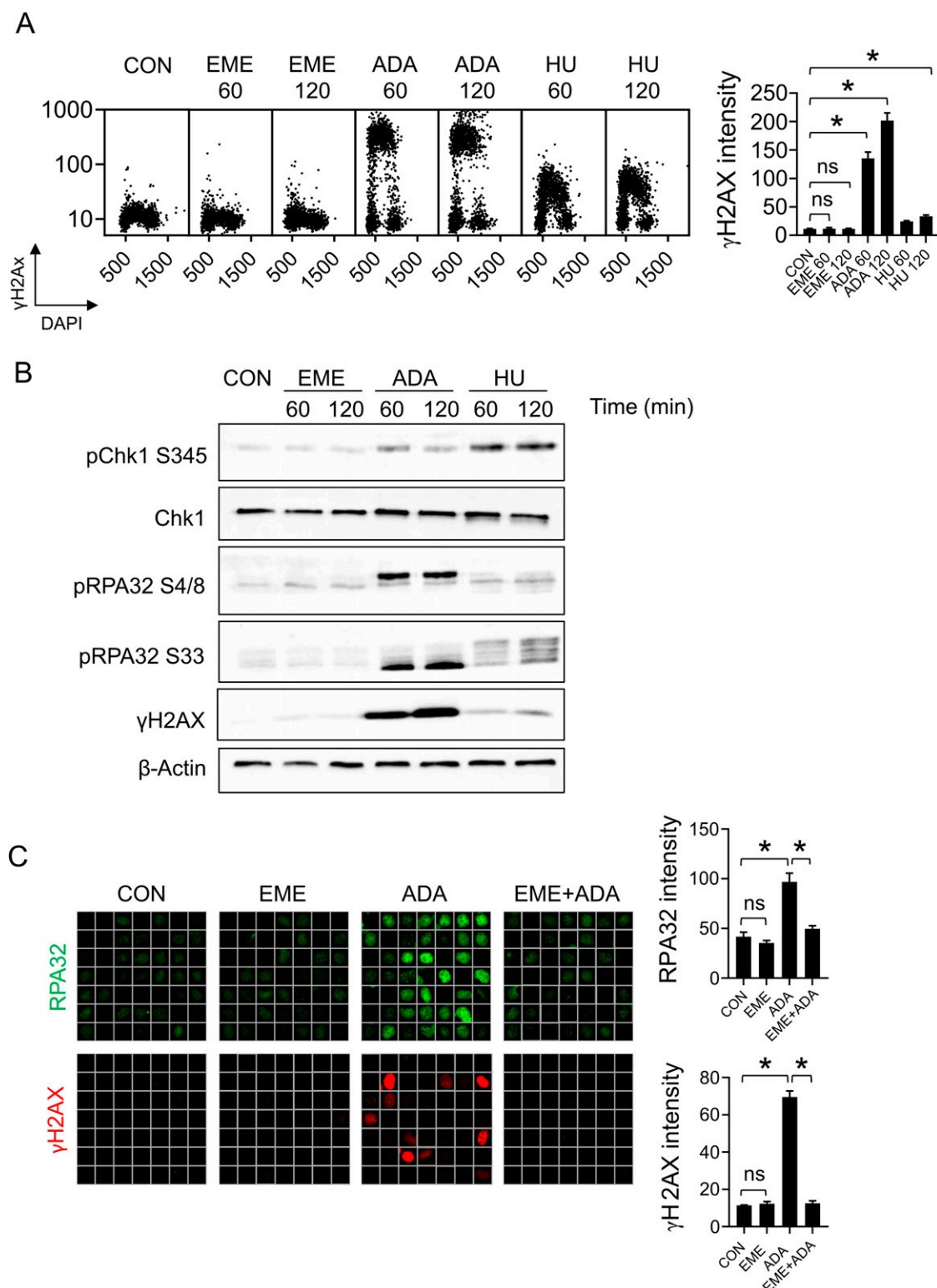

**Figure 3. Emetine does not activate the replication checkpoint or DNA damage response.**
**(A)** U2OS cells were treated with 1 μM EME, 1 μM ADA, or 2 mM HU for 60 or 120 min and stained for γH2AX. The bar graph shows mean γH2AX intensities ± SD from three independent experiments (n = 3). Statistical analysis by one-way ANOVA. *P < 0.05; ns, nonsignificant. **(B)** U2OS cells were treated as in (A) and cell lysates were analyzed by immunoblotting with indicated antibodies. **(C)** Representative ScanR images (left) and quantification (right) of mean CB-RPA32 (top) and γH2AX (bottom) in U2OS treated with EME, ADA, or a combination of both. Where indicated, U2OS cells were incubated with 1 μM EME for 20 min before treatment with 1 μM ADA for an additional 20 min. The bar graphs show mean CB-RPA32 and γH2AX intensities ± SD from three independent experiments (n = 3). Statistical analysis by one-way ANOVA. *P < 0.05; ns, nonsignificant.

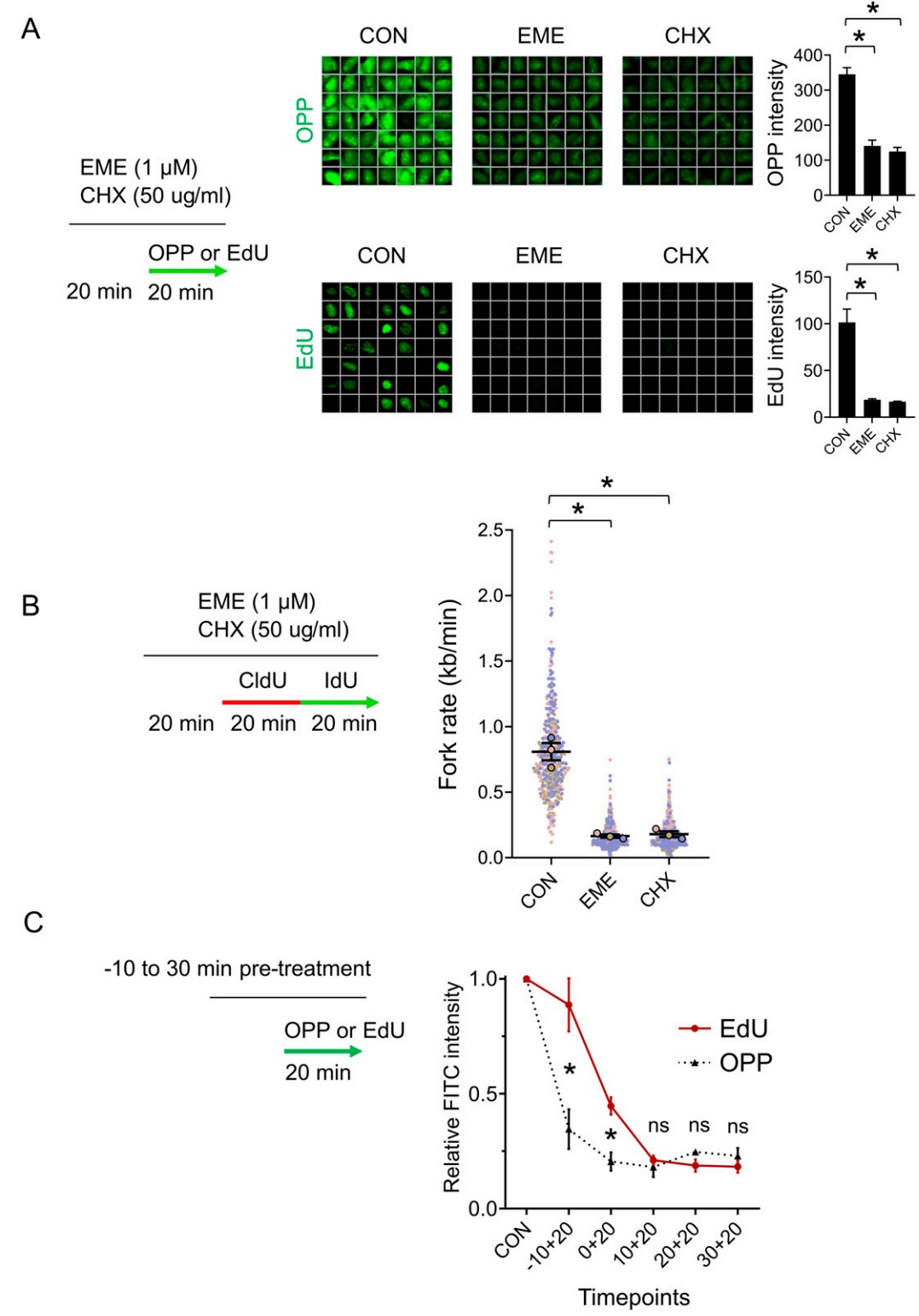

**Figure 4. Emetine's anti-proteosynthetic activity blocks DNA replication.**
**(A)** Representative ScanR images (left) and quantification (right) of mean OPP (top) and EdU (bottom) signal in U2OS cells pretreated with 1 μM EME or 50 μg/ml CHX for 20 min, treated with either 10 μM EdU or 10 μM OPP for 20 min and stained for OPP or incorporated EdU by click chemistry. The bar graphs show mean OPP and EdU intensities ± SD from three independent experiments (n = 3). Statistical analysis by one-way ANOVA. *$P < 0.05$. **(B)** U2OS cells were pretreated with 1 μM EME or 50 μg/ml CHX for 20 min, labeled with CldU/IdU, and fork rates were determined by DNA combing according to the length of the IdU track. The superplot shows results from three independent experiments (n = 3, blue, pink and brown). Circles show mean values of each experiment containing 150 DNA fibers. The average of mean values from each

supporting the hypothesis that EME blocks replication on leading and lagging strands. The third criterion is that extensive strand uncoupling is associated with activation of replication checkpoint, DNA damage response, and eventually replication catastrophe. Even prolonged EME treatment, in contrast to ADA or general inhibition of DNA replication by HU, did not increase markers of replication checkpoint or DNA damage. Overall, the data presented in this work suggest that EME is not a specific inhibitor of lagging strand synthesis. We propose that EME's inhibition of proteosynthesis, which temporally precedes its effects on DNA replication, is the mechanism of EME's action toward DNA replication likely through the deprivation of nascent histones.

Emetine was used in the original protocol for mapping the origins of bidirectional replication units (Handeli et al, 1989) based on the assumption of conservative nucleosome segregation in the absence of histone synthesis, leaving nascent DNA on the retrograde arm unprotected by histones and sensitive to non-specific endonucleases. When Burhans and colleagues reproduced this protocol, they found that in presence of EME nuclease digestion is not required and histones segregate randomly to both DNA strands (Burhans et al, 1991). To explain their data, they proposed a model of imbalanced DNA synthesis via preferential inhibition of OF synthesis. Two observations supported the hypothesis that EME preferentially inhibits OF synthesis. First, a fraction of RNA-primed OFs was rapidly diminished, and second, ssDNA was detected on one arm of replication forks after EME treatment. Burhans and colleagues (Burhans et al, 1991) showed that not only OFs but also the synthesis of long nascent DNA were dramatically reduced in the presence of EME. Moreover, the authors (Burhans et al, 1991) even stated that "emetine eventually reduced the overall rate of DNA synthesis to less than 10% of controls" in their discussion. Such a reduction in DNA synthesis is consistent with our data. Another criterion for the specific inhibition of OF synthesis is the continuous generation of ssDNA at the lagging strand. Burhans and colleagues did indeed show the presence of short ssDNA on one arm of the replication forks after EME treatment and considered that as the evidence supporting the specific inhibition of lagging strand synthesis by EME. It is important to note that the length of ssDNA after 20 and 40 min of EME treatment was indistinguishable (Burhans et al, 1991). Therefore, after only 20 min of EME treatment, DNA replication on both strands either continued at the same reduced speed or had already stalled. However, neither of those two options is consistent with specific inhibition of lagging strand synthesis. On the other hand, the authors (Burhans et al, 1991) did not propose the use of EME as a specific inhibitor of lagging strand synthesis, rather they clarified a model for the identifying origins of bidirectional DNA replication. EME has been misused for this purpose decades later. For example, EME was used to show that PARP activity during S-phase is prevented by suppressing OF formation

(Hanzlikova et al, 2018). Considering our data, reduced PARylation after EME treatment reflects a general block in DNA replication rather than a specific inhibition of lagging strand synthesis. Nevertheless, here we show that suppression of lagging strand synthesis via mild POLA1 inhibition leads to a reduction of PARylation induced by PARG inhibition or LIG1 knockdown, furthering the intermediates of OF processing as sources of S-phase PARylation.

In conclusion, we provide evidence that emetine is not a specific inhibitor of lagging strand synthesis. Emetine rapidly and fully inhibits DNA replication on both strands. Importantly, unlike inhibitors of POLA, Emetine does not induce strand uncoupling and activation of the replication checkpoint. From a broader perspective, our study further contributes to understanding the relationship between PARP metabolism and OF processing during DNA replication.

# Materials and Methods

### Cell lines

Human osteosarcoma U2OS and diploid retinal pigment epithelium RPE1 cells were grown in DMEM (LM-D1110/500; Biosera) supplemented with 10% fetal bovine serum (10270106; Gibco) and penicillin/streptomycin (P4333; Sigma-Aldrich). All cell types were purchased from ATCC and were regularly tested for mycoplasma contamination.

### Chemicals

In some experiments, cells were treated with the following drugs: olaparib (S1060; Selleck Chemicals), emetine dihydrochloride (E0100000; Sigma-Aldrich) - EME, adarotene (HY-14808; MedChemExpress) - ADA, CD437 (C5865; Sigma-Aldrich) - CD, hydroxyurea (H8627; Sigma-Aldrich) - HU, cycloheximide (C4859; Sigma-Aldrich) - CHX, and PARG inhibitor (PDD 0017273; Tocris) - —PARGi.

### Antibodies

PAR (MABE1031; Millipore)
RPA32 (ab2175; Abcam)
BrdU (RPN20AB; AP Biotech)
γH2AX (05-636; Millipore)
phosphoChk1 S345 (#2348; Cell Signaling)
Chk1 (sc-8408; Santa Cruz Biotechnology)
phosphoRPA32 S4/8 (A300-245A; Bethyl Laboratories)
phosphoRPA32 S33 (A300-246A; Bethyl Laboratories)
β-Actin (sc-47778; Santa Cruz Biotechnology)

---

experiment (black horizontal bar) with standard error of the mean (black error bars) are indicated. Mean values from each experiment were statistically tested by paired two-tailed *t* test. *$P < 0.05$. **(C)** U2OS cells were pretreated with 1 $\mu$M EME or 50 $\mu$g/ml CHX for the indicated time, treated with either 10 $\mu$M EdU or 10 $\mu$M OPP for the last 20 min, and stained for OPP or incorporated EdU by click chemistry. The graph shows mean OPP and EdU intensities ± SD from three independent experiments (n = 3). Statistical analysis by two-tailed *t* test. *$P < 0.05$; ns, nonsignificant.

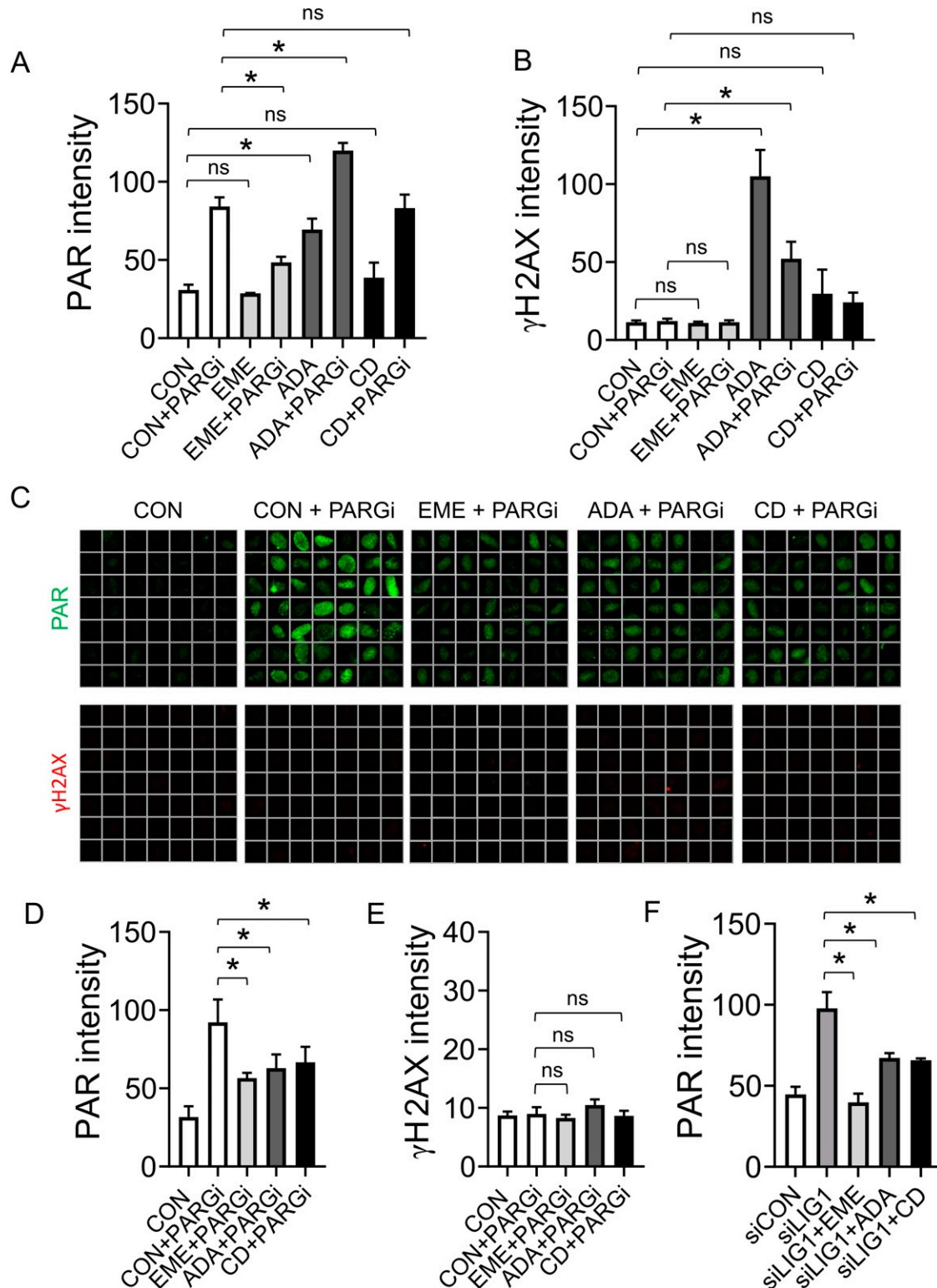

**Figure 5. PARP activity during S-phase is prevented by mild inhibition of POLA1.**
**(A)** U2OS cells were treated with 1 $\mu$M EME, 1 $\mu$M ADA, or 2 $\mu$M CD437 for 40 min, and with 10 $\mu$M PARG inhibitor for the last 20 min. PAR signal was detected using indirect immunofluorescence and quantified. The bar graph shows mean PAR intensities ± SD from three independent experiments (n = 3). Statistical analysis by one-way ANOVA. *$P$ < 0.05; ns, nonsignificant. **(A, B)** Quantification of $\gamma$H2AX signal in U2OS cells treated as in (A). The bar graph shows mean $\gamma$H2AX intensities ± SD from three independent experiments (n = 3). Statistical analysis by one-way ANOVA. *$P$ < 0.05; ns, nonsignificant. **(C)** Representative ScanR images of PAR and $\gamma$H2AX signals in U2OS cells treated with 100 nM ADA or 200 nM CD437 for 16 h, and with 10 $\mu$M PARG inhibitor for the last 20 min. **(C, D)** Quantification of PAR signal in cells from (C). The bar graph shows mean

## RNA interference

All siRNA transfections were performed using Lipofectamine RNAiMAX (13778075; Invitrogen) according to the manufacturer's instructions. Unless specified otherwise, siRNAs were obtained from Ambion as Silencer Select reagents and used at a final concentration of 14 nM. All experiments were performed 72 h after transfection.

siCON (negative control #1, AM4635, 5′-AGUACUGCUUACGAUACGGTT-3′)
siLIG1 (s8174, 5′-GGAUCCAUCUGGUUACAAUtt-3′)

## DNA combing

DNA combing was performed as previously described (Moudry et al, 2022). In brief, cells were labeled with 25 µM CldU (I7125; Sigma-Aldrich) and 250 µM IdU (C6891; Sigma-Aldrich) for 20 min. DNA was then extracted using a FiberPrep kit (EXT-001; Genomic Vision) following the manufacturer's instructions. Extracted DNA was combed on vinylsilane coated CombiCoverslips (COV-002-RUO; Genomic Vision), denatured, dehydrated, air-dried and blocked. Coverslips were incubated with primary antibodies, mouse anti-BrdU (1:10, BD347580; BD Biosciences), and rat anti-BrdU (1:50, ab6326; Abcam) antibodies. After four washes with PBS, cover glasses were incubated with secondary antibodies goat anti-mouse Alexa Fluor 488 (1:100, A11001; Invitrogen) and goat anti-rat Alexa Fluor 568 (1:100, ab175476; Abcam) antibodies. After four washes with PBS, cover glasses were air-dried and mounted using Vecta-shield (H-1000; Vector Laboratories). Images of DNA fibers were acquired using CellObserver spinning disc confocal microscopic system (Zeiss), and length of labeled DNA was analyzed using ImageJ software.

## Immunoblotting

Cells were grown in 60-mm cell culture dishes and whole-cell extracts were obtained by lysis in Laemmli sample buffer (50 mM Tris–HCl [pH 6.8], 100 mM DTT, 2.0% SDS, 0.1% bromophenol blue, 10% glycerol) and analyzed by SDS–polyacrylamide gel electrophoresis following standard procedures. Primary antibodies were incubated overnight at 4°C in TBS-Tween 20 containing 5% powder milk. Secondary HRP-coupled antibodies (NA931 and NA934; GE Healthcare) were incubated at room temperature for 1 h. Chemiluminescence was detected with a ChemiDoc XRS+ imaging system (Bio-Rad).

## Immunofluorescence

Immunofluorescence staining was performed as previously described (Frankum et al, 2015). Shortly, cells grown on 12 mm wide glass coverslips (41001112; Assistent) were washed twice in PBS, fixed for 15 min at RT with 4% formaldehyde, washed in PBS, permeabilized for 5 min with 0.2% Triton X-100 in PBS, then washed again in PBS before being incubated with primary antibodies for 60 min at RT. After the washing step, the coverslips were incubated with goat anti-rabbit Alexa Fluor 488 or goat anti-mouse Alexa Fluor 568 secondary antibodies (A11034 and A11004; Invitrogen) for 60 min at RT, then washed with PBS, and finally mounted using Vectashield mounting medium with DAPI (H-1200; Vector Laboratories).

For detection of CB-RPA32 pre-extraction was carried out before fixation by incubating the cells in 0.2% Triton X-100 PBS on ice for 5 min.

For detection of ssDNA, cells were grown on coverslips in culture media with 10 µM BrdU for 48 h before indicated treatment. After treatment, cells were fixed in ice-cold methanol for 20 min at 4°C and washed with a 1:1 mixture of methanol/acetone, followed by a regular immunofluorescence staining using BrdU antibody (RPN20AB; AP Biotech).

For detection of nascent DNA synthesis, cells were incubated with 10 µM EdU 20 min before fixation and EdU detection was performed using Click-iT EdU Alexa Fluor 594 imaging kit (C10639; Invitrogen) according to the manufacturer's recommendations.

For detection of protein synthesis, cells were incubated with 10 µM OPP 20 min before fixation and OPP detection was performed using Click-iT Plus OPP Alexa Fluor 488 (C10456; Invitrogen) according to the manufacturer's recommendations.

## Microscope image acquisition

Quantitative image-based cytometry (QIBC) of the immunofluorescence-stained samples was performed using an automatic inverted fluorescence microscope BX71 (Olympus) using ScanR acquisition software (Olympus) and analyzed with ScanR analysis software (Olympus).

## Clonogenic assay

Sensitivity to ADA or CD437 was determined by plating 100 U2OS cells on a 12-well plate. Colonies were allowed to grow for 8 d, fixed in 70% ethanol, and stained with 0.5% crystal violet in 20% methanol. Colonies were counted, and the surviving fraction was calculated and normalized to untreated control.

# Supplementary Information

---

PAR intensities ± SD from three independent experiments (n = 3). Statistical analysis by one-way ANOVA. *P < 0.05. **(C, E)** Quantification of γH2AX signal in cells from (C). The bar graph shows mean γH2AX intensities ± SD from three independent experiments (n = 3). Statistical analysis by one-way ANOVA. *P < 0.05; ns, nonsignificant. **(F)** Quantification of PAR signal in U2OS cells transfected with indicated siRNAs in the presence of EME (1 µM, 20 min), ADA (100 nM, 16 h), or CD437 (200 nM, 16 h). The bar graph shows mean PAR intensities ± SD from three independent experiments (n = 3). Statistical analysis by one-way ANOVA. *P < 0.05.

## Acknowledgements

This project was supported by the Czech Science Foundation (grant no. 20-03457Y), the MEYS CR (Large RI Project LM2018129 - Czech-BioImaging) and the project National Institute for Cancer Research (Programme EXCELES, ID Project No. LX22NPO5102) - Funded by the European Union - Next Generation EU.

### Author Contributions

D Lukac: investigation, visualization, and writing—review and editing.
Z Machacova: investigation and writing—review and editing.
P Moudry: conceptualization, supervision, funding acquisition, investigation, writing—original draft, and project administration.

### Conflict of Interest Statement

The authors declare that they have no conflict of interest.

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
