## [Reviewer comments · Life Science Alliance]

Life Science Alliance

Emetine blocks DNA replication via proteosynthesis inhibition not by targeting Okazaki fragments

David Lukac, Zuzana Machacova, and Pavel Moudry

DOI: <https://doi.org/10.26508/lsa.202201560>

Corresponding author(s): *Pavel Moudry, Institute of Molecular and Translational Medicine*

Review Timeline:

Submission Date:	2022-06-14
Editorial Decision:	2022-07-18
Revision Received:	2022-08-25
Editorial Decision:	2022-08-29
Revision Received:	2022-08-30
Accepted:	2022-08-30

Scientific Editor: Novella Guidi

Transaction Report:

July 18, 2022

Re: Life Science Alliance manuscript #LSA-2022-01560-T

Dr. Pavel Moudry
Institute of Molecular and Translational Medicine
Hnevotinska 5
Olomouc 77900
Czech Republic

Dear Dr. Moudry,

Thank you for submitting your manuscript entitled "Emetine blocks DNA replication via proteosynthesis inhibition not by targeting Okazaki fragments" to Life Science Alliance. The manuscript was assessed by expert reviewers, whose comments are appended to this letter. We invite you to submit a revised manuscript addressing the Reviewer comments.

Thank you for this interesting contribution to Life Science Alliance. We are looking forward to receiving your revised manuscript.

Sincerely,

B. MANUSCRIPT ORGANIZATION AND FORMATTING:

Reviewer #1 (Comments to the Authors (Required)):

The work by Lukac and colleagues clearly demonstrates that emetin does not specifically inhibit lagging strand synthesis during DNA replication. Moreover, this work confirms by using the POLA1 inhibitor adarotene that S-phase PARylation is linked to the processing of Okazaki fragments during lagging strand synthesis.

The experiments are well designed and controlled throughout the study and I have no concerns preventing the publication of their manuscript in Life Science Alliance.

I congratulate the authors for this clarifying work that will be important to avoid the inappropriate use of emetin in future studies.

Minor comments:

I would advise the authors to mention their results about S-phase PARylation in the abstract for the community to reach them more easily, and preventing the sole reference to Hanzlikova et al., 2018.

Regarding the representation and statistical analysis of the DNA combing data, it is not clear to me if the figures represent only one replicate or the sum of the 3 replicates.

I would advise the authors to refer to Lord et al., JCB 2020 to build superplots and/or to perform statistical analyses between mean values of each experiment with a paired t test or Anova.

Reviewer #2 (Comments to the Authors (Required)):

This manuscript described the inhibition of DNA replication by an alkaloid emetine and the mechanism. Emetine is an inhibitor of protein biosynthesis, whereas emetine was used as a specific inhibitor of lagging strand synthesis which prevents the formation of Okazaki fragments, thereby resulting in uncoupling leading and lagging strand replication for 20 years. The authors show that emetine inhibit DNA replication of both strands and it is result from inhibition of protein synthesis. This study reveals that emetine is not a specific lagging strand synthesis inhibitor. The reviewer considers that this study provides new significant findings about physiological activity of emetine. Based on this study, researchers of molecular biology will understand the activity of emetine and they will recognize that they should use emetine for appropriate purposes in their future researches. The experiments were well designed and demonstrated. The manuscript was well organized and concisely written. Therefore, the reviewer has several concerns or comments, as follows.

Major points

(1) The authors described that short incubation with POLA1 inhibitor did not suppress PARGi-induced PARylation. In contrast, mild POLA1 inhibition suppressed PARGi-induced PARylation to similar levels as EME treatment. Also, mild POLA1 inhibition also partially suppressed PARylation induced by LIG1 knockdown, further supporting the model where unligated OFs contribute to PARylation.

The reviewer could not understand that explanation. It would be helpful for general readers if the authors would provide clearer explanation.

(2) The effects of ADA on PAR and gH2AX intensities seem to be larger than those of CD437 in Figs. 5A and 5B. The reviewer appreciates if the authors provide comments on difference of those POLA1 inhibitors.

Minor points

(1) In chemicals, the authors should add CD437. The reviewer recommends that abbreviations of chemicals, such as EME, ADA, ..., are also shown here.

(2) In immunofluorescence for protein synthesis, "EU detection" would be "OPP detection". Also, the authors detected nascent DNA strands by EdU using Click-iT kit. The authors should also describe that experiment.

(3) The authors describe that POLA1 is the primase subunit of POLA complex somewhere.

Reviewer #3 (Comments to the Authors (Required)):

In this paper Lukac and colleagues report on a study of the effect of emetine on DNA replication. Emetine, a plant-derived alkaloid, has been accepted as a DNA synthesis inhibitor, specific for the lagging strand. The authors use a combination of techniques, including DNA combing, RNAi, immunoblotting, immunofluorescence to measure the effect of the drug on DNA replication and protein synthesis in two human cell lines, osteosarcoma and retinal pigment epithelium cells. The authors show that while emetine completely blocks DNA replication, in contrast to adarotene, a POLA1 inhibitor, it does not generate ssDNA or increase RPA binding to chromatin. Emetine, unlike adarotene and hydroxyurea, does not activate the replication checkpoint or markers of DNA damage. The authors also observe that inhibition of protein synthesis by emetine occurs at an earlier timepoint than its effect on DNA replication.

Based on these results the authors conclude that emetine is not a lagging strand synthesis-specific inhibitor, rather it blocks synthesis of both DNA strands by inhibition of proteosynthesis. The results are well presented and support the authors' conclusions. The paper is clearly written and it is likely to be of significant interest to those working on DNA replication. I have only several minor points that may improve the quality of the presentation.

1. In the Abstract the authors state: " ..we demonstrate that the acute block of proteosynthesis by emetine temporally precedes its effects on DNA replication. Thus, by inhibiting proteosynthesis, emetine affects DNA replication." The hypothesis that inhibition of DNA synthesis by emetine is a consequence of protein synthesis inhibition by the drug is reasonable. However, the effect of emetine on specific proteins involved in replication was not investigated directly. Perhaps in the Abstract and throughout the paper the authors should use more cautious language. For example, the results are consistent with the hypothesis that.....

2. If I'm not mistaken, emetine is not mentioned in the reference Thakar et al. (page 3, top)

3. The figure numbers are missing

4. The pages of the manuscript are not numbered

5. Page 11, line 8 from bottom, "....that both EME and CHX firstly block...." Should be " first block."

6. Page 8, "Quantitative microscopy-based cytometry (QIBC)", shouldn't it be quantitative image-based cytometry ?

(Our replies are in a standard format and the *Reviewer's comments are in italics and blue*)

Reviewer #1 (Comments to the Authors (Required)):

The work by Lukac and colleagues clearly demonstrates that emetin does not specifically inhibit lagging strand synthesis during DNA replication. Moreover, this work confirms by using the POLA1 inhibitor adarotene that S-phase PARylation is linked to the processing of Okazaki fragments during lagging strand synthesis.

The experiments are well designed and controlled throughout the study and I have no concerns preventing the publication of their manuscript in Life Science Alliance.

I congratulate the authors for this clarifying work that will be important to avoid the inappropriate use of emetin in future studies.

We really appreciate the positive remarks of this Reviewer, and we are also grateful for insightful comments.

Minor comments:

I would advise the authors to mention their results about S-phase PARylation in the abstract for the community to reach them more easily, and preventing the sole reference to Hanzlikova et al., 2018.

As advised by the Reviewer we added results about S-phase PARylation in the abstract.

Regarding the representation and statistical analysis of the DNA combing data, it is not clear to me if the figures represent only one replicate or the sum of the 3 replicates.

I would advise the authors to refer to Lord et al., JCB 2020 to build superplots and/or to perform statistical analyses between mean values of each experiment with a paired t test or Anova.

DNA combing data were presented by one replicate in the previous version of the figures. We modified all figures containing DNA combing results according to Lord et al., JCB 2020 and generated superplots showing 3 independent experiments (n=3), each experiment with 150 analyzed replication forks. Mean values of independent experiments are statistically tested by paired two-tailed *t* test, as suggested by this Reviewer.

Reviewer #2 (Comments to the Authors (Required)):

This manuscript described the inhibition of DNA replication by an alkaloid emetine and the mechanism. Emetine is an inhibitor of protein biosynthesis, whereas emetine was used as a specific inhibitor of lagging strand synthesis which prevents the formation of Okazaki fragments, thereby resulting in uncoupling leading and lagging strand replication for 20 years. The authors show that emetine inhibit DNA replication of both strands and it is result from inhibition of protein synthesis. This study reveals that emetine is not a specific lagging strand synthesis inhibitor. The reviewer considers that this study provides new significant findings about physiological activity of emetine. Based on this study, researchers of molecular biology will understand the activity of emetine and they will recognize that they should use emetine for appropriate purposes in their future researches. The experiments were well designed and demonstrated. The manuscript was well organized and concisely written. Therefore, the reviewer has several concerns or comments, as follows.

Major points

(1) The authors described that short incubation with POLA1 inhibitor did not suppress PARGi-induced PARylation. In contrast, mild POLA1 inhibition suppressed PARGi-induced PARylation to similar levels as EME treatment. Also, mild POLA1 inhibition also partially suppressed PARylation induced by LIG1 knockdown, further supporting the model where unligated OFs contribute to PARylation.

The reviewer could not understand that explanation. It would be helpful for general readers if the authors would provide clearer explanation.

Acute inhibition of POLA1 (short incubation with μM concentrations of POLA1 inhibitors) by itself (without PARGi) induces an increase in PARylation (Figure 5A) and γH2Ax (Figure 5B) signals and leads to replication catastrophe (Ercilla et al., Cell Rep 2020). Therefore, this treatment is unsuitable for testing whether PARGi-induced PARylation is dependent on POLA1 activity.

Mild POLA1 inhibition (long incubation with 100 - 200 nM POLA1 inhibitors) does not induce γH2Ax signal (Figure 5C-E), but still shows markers of strand uncoupling (Supplementary Figure 5A-C) and is tolerated by U2OS cells (Supplementary Figure 5D and Ercilla et al., Cell Rep 2020). We show that both PARGi- or LIG1 knock-down- induced PARylation are reduced using mild POLA1 inhibition, suggesting that S-phase PARylation is linked to Okazaki fragment processing during lagging strand synthesis. We modified this section of the results to be clearer for general readers.

(2) The effects of ADA on PAR and γ H2AX intensities seem to be larger than those of CD437 in Figs. 5A and 5B. The reviewer appreciates if the authors provide comments on difference of those POLA1 inhibitors.

Indeed, ADA is a more potent POLA1 inhibitor than CD437 in the induction of PARylation (Figure 5A) and γ H2Ax (Figure 5B). This was also observed earlier by Ercilla and colleagues (Ercilla et al., Cell Rep 2020). We modified the results and mentioned this point in the text.

Minor points

(1) In chemicals, the authors should add CD437. The reviewer recommends that abbreviations of chemicals, such as EME, ADA, ..., are also shown here.

Thank you, CD437 is added in the chemicals section. Abbreviations of chemicals used throughout the manuscript are also added in the chemicals section.

(2) In immunofluorescence for protein synthesis, "EU detection" would be "OPP detection". Also, the authors detected nascent DNA strands by EdU using Click-iT kit. The authors should also describe that experiment.

We apologize for the mistakes in the methods. We corrected the immunofluorescence section as suggested by the Reviewer.

(3) The authors describe that POLA1 is the primase subunit of POLA complex somewhere.

We thank Reviewer for this comment, we modified the introduction and describe subunits of POLA complex there.

Reviewer #3 (Comments to the Authors (Required)):

In this paper Lukac and colleagues report on a study of the effect of emetine on DNA replication. Emetine, a plant-derived alkaloid, has been accepted as a DNA synthesis inhibitor, specific for the lagging strand. The authors use a combination of techniques, including DNA combing, RNAi, immunoblotting, immunofluorescence to measure the effect of the drug on DNA replication and protein synthesis in two human cell lines, osteosarcoma and retinal pigment epithelium cells. The authors show that while emetine completely blocks DNA replication, in contrast to adarotene, a POLA1 inhibitor, it does not generate ssDNA or increase RPA binding to chromatin. Emetine, unlike adarotene and hydroxyurea, does not activate the replication checkpoint or markers of DNA damage. The authors also observe that inhibition of protein synthesis by emetine occurs at an earlier timepoint than its effect on DNA replication.

Based on these results the authors conclude that emetine is not a lagging strand synthesis-specific inhibitor, rather it blocks synthesis of both DNA strands by inhibition of proteosynthesis. The results are well presented and support the authors' conclusions. The paper is clearly written and it is likely to be of significant interest to those working on DNA replication.

I have only several minor points that may improve the quality of the presentation.

1. In the Abstract the authors state: " ..we demonstrate that the acute block of proteosynthesis by emetine temporally precedes its effects on DNA replication. Thus, by inhibiting proteosynthesis, emetine affects DNA replication." The hypothesis that inhibition of DNA synthesis by emetine is a consequence of protein synthesis inhibition by the drug is reasonable. However, the effect of emetine on specific proteins involved in replication was not investigated directly. Perhaps in the Abstract and throughout the paper the authors should use more cautious language. For example, the results are consistent with the hypothesis that.....

We agree with the Reviewer that we did not test the effects of emetine on specific protein(s). We have reformulated this in the abstract and results.

2. If I'm not mistaken, emetine is not mentioned in the reference Thakar et al. (page 3, top)

We do not agree on this point with the Reviewer. Emetine was used by Thakar et al., to confirm that PAR chains in KR cells are caused by the accumulation of unligated Okazaki fragments. These results were probably missed by the Reviewer because they are not in the main figures but in the supplementary figure 6B:

" Confirming that PAR chains are caused by accumulation of unligated OFs, their formation was suppressed upon short treatment with emetine, an inhibitor of lagging strand synthesis which prevents formation of OFs, uncoupling leading and lagging strand replication⁴⁹ (Supplementary Fig. 6b)."

3. The figure numbers are missing

Thank you, this is now corrected.

4. The pages of the manuscript are not numbered

Thank you, page numbers are added.

5. Page 11, line 8 from bottom, "...that both EME and CHX firstly block...."Should be " first block."

Thank you, this is now corrected.

6. Page 8, "Quantitative microscopy-based cytometry (QIBC)", shouldn't it be quantitative image-based cytometry ?

Thank you, this is now corrected to "Quantitative image-based cytometry".

August 29, 2022

RE: Life Science Alliance Manuscript #LSA-2022-01560-TR

Dr. Pavel Moudry
Institute of Molecular and Translational Medicine
Hnevotinska 5
Olomouc 77900
Czech Republic

Dear Dr. Moudry,

Thank you for submitting your revised manuscript entitled "Emetine blocks DNA replication via proteosynthesis inhibition not by targeting Okazaki fragments". We would be happy to publish your paper in Life Science Alliance pending final revisions necessary to meet our formatting guidelines.

- please add the Twitter handle of your host institute/organization as well as your own or/and one of the authors in our system
- please consult our manuscript preparation guidelines <https://www.life-science-alliance.org/manuscript-prep> and make sure your manuscript sections are in the correct order
- please use the [10 author names, et al.] format in your references (i.e. limit the author names to the first 10)

A. FINAL FILES:

B. MANUSCRIPT ORGANIZATION AND FORMATTING:

**Submission of a paper that does not conform to Life Science Alliance guidelines will delay the acceptance of your

manuscript.**

The license to publish form must be signed before your manuscript can be sent to production. A link to the electronic license to publish form will be sent to the corresponding author only. Please take a moment to check your funder requirements.

Sincerely,

August 30, 2022

RE: Life Science Alliance Manuscript #LSA-2022-01560-TRR

Dr. Pavel Moudry
Institute of Molecular and Translational Medicine
Hnevotinska 5
Olomouc 77900
Czech Republic

Dear Dr. Moudry,

Thank you for submitting your Research Article entitled "Emetine blocks DNA replication via proteosynthesis inhibition not by targeting Okazaki fragments". It is a pleasure to let you know that your manuscript is now accepted for publication in Life Science Alliance. Congratulations on this interesting work.

DISTRIBUTION OF MATERIALS:

Again, congratulations on a very nice paper. I hope you found the review process to be constructive and are pleased with how the manuscript was handled editorially. We look forward to future exciting submissions from your lab.

Sincerely,
